# Epigenetic Variation at a Genomic Locus Affecting Biomass Accumulation under Low Nitrogen in *Arabidopsis thaliana*

**Markus Kuhlmann [1],\*,†** , **Rhonda C. Meyer [1],\*,†** , **Zhongtao Jia [2],†** , **Doreen Klose [1],†** , **Lisa-Marie Krieg [1],†** , **Nicolaus von Wirén [2],†** and **Thomas Altmann [1],†**

[1]   Heterosis, Department of Molecular Genetics, Leibniz Institute of Plant Genetics and Crop Plant Research (IPK), OT Gatersleben, 06466 Seeland, Germany; doreen.klose@student.uni-halle.de (D.K.); Krieg@ipk-gatersleben.de (L.-M.K.); altmann@ipk-gatersleben.de (T.A.)

[2]   Molecular Plant Nutrition, Department of Physiology and Cell Biology, Leibniz Institute of Plant Genetics and Crop Plant Research (IPK), OT Gatersleben, 06466 Seeland, Germany; zhongtao@ipk-gatersleben.de (Z.J.); vonwiren@ipk-gatersleben.de (N.v.W.)

\*   Correspondence: kuhlmann@ipk-gatersleben.de (M.K.); meyer@ipk-gatersleben.de (R.C.M.)

†   Authors contribute equally to this work.

**Abstract:** Nitrogen (N) is a macronutrient determining crop yield. The application of N fertilisers can substantially increase the yield, but excess use also causes the nitrate pollution of water resources and increases production costs. Increasing N use efficiency (NUE) in crop plants is an important step to implement low-input agricultural systems. We used *Arabidopsis thaliana* as model system to investigate the natural genetic diversity in traits related to NUE. Natural variation was used to study adaptive growth patterns and changes in gene expression associated with limited nitrate availability. A genome-wide association study revealed an association of eight SNP markers on Chromosome 1 with shoot growth under limited N. The identified linkage disequilibrium (LD) interval includes the DNA sequences of three cysteine/histidine-rich C1 domain proteins in tandem orientation. These genes differ in promoter structure, methylation pattern and expression level among accessions, correlating with growth performance under N deficiency. Our results suggest the involvement of epigenetic regulation in the expression of NUE-related traits.

**Keywords:** biomass on low N; DNA-methylation; cysteine/histidine-rich C1 domain proteins

## 1. Introduction

Nitrogen ($N_2$) is the most abundant element in the Earth's atmosphere (78%) but is sparsely distributed in the soil, making soil nitrogen a growth limiting factor for plants. Nitrogen is also an elementary building block for many essential molecules required for life: as part of amino acids, it contributes to protein synthesis, and in nucleic acids, it contributes to the conservation and transcription of genetic information. Furthermore, it is a key element in chlorophyll, the active core of photosynthesis [1]. Consequently, plant growth depends on a continuous uptake of N. In agricultural plant production, most of the N required by plants needs to be provided by fertilisation. However, the extensive use of fertilisers leads to N leakage, causing the eutrophication of surface waters or contamination of ground and drinking water with nitrate [2]. It is therefore of great importance to reduce N fertiliser input, and one way is to improve nitrogen use efficiency (NUE) in plants to breed crop varieties with stable yields under reduced N fertilisation.

We exploited the model plant system, *Arabidopsis thaliana*, to identify genes that can serve as tools to improve crop nitrogen use efficiency (NUE). Several important genes and regulatory factors

involved in nitrogen uptake, assimilation and utilisation have been identified in Arabidopsis [3–5]. Recently, the molecular basis for NUE was addressed in five contrasting Arabidopsis accessions grown under different concentrations of inorganic N [6]. In these hydroponic experiments, the amount of nitrate found in roots and shoots was correlated to the expression of genes involved in nitrogen uptake and assimilation in the root. Plant responses to low N span various levels of gene regulation, including differential gene expression, post-transcriptional regulation by miRNAs (miR-167, miR-169 and miR-393) or the modulation of enzyme activities [7]. In an earlier approach, the root-specific expression of the RNA-directed RNA-polymerase RDR2 was found to be negatively correlated with biomass accumulation under N-limiting conditions [8]. As RDR2 is involved in epigenetic regulation via the generation of double-stranded RNA and RNA-directed DNA methylation, we wanted to analyse the involvement of this regulatory process in plant growth under varying nitrogen levels.

Genome-wide association studies (GWAS) have contributed substantially to the identification of genes and the elucidation of the regulatory pathways involved in N-dependent growth processes or NUE [9,10].

Here, we took advantage of the availability of the large dataset of biomass data of Arabidopsis plants grown under limiting N conditions [8] to perform a genome-wide association study (GWAS) that identified a region on Chromosome 1. We investigated the growth responses of selected Arabidopsis accessions with differing methylation patterns in quantitative trait loci (QTL) candidate genes to N deficiency in a dual approach, growing accessions both on soil or on agar plates under sufficient and low N supply to monitor shoot biomass.

## 2. Materials and Methods

### 2.1. Biological Materials

The Arabidopsis collections were described in [8]. Briefly, the two complementary populations consist of 102 Arabidopsis accessions (Table S1) and of 123 recombinant inbred lines (mRILs) assembled from nine crosses (Table S2). The accessions represent a wide range of geographic origins (Figure 1).

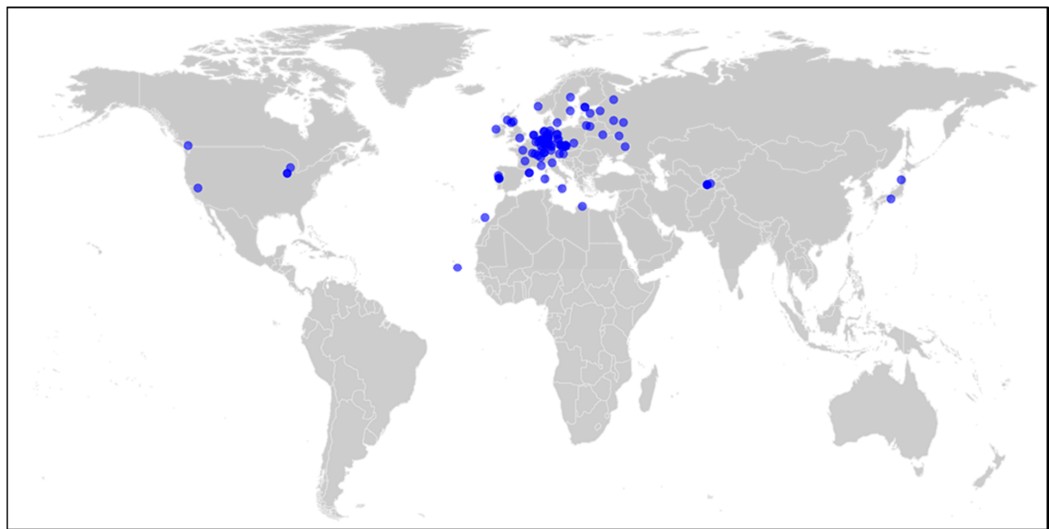

**Figure 1.** Geographic origin of the 102 Arabidopsis accessions used in this study. Each blue dot indicates the origin of one accession.

For the analysis of epigenetic effects in the QTL, the following homozygous T-DNA insertion lines were used: *rdr2-1* (SAIL1277_H08) [11], *rdr6-11* [12], *suvh2* (Gabi_Kat_516A07), *suvh9* (SALK_048033) and *suvh2/9* double mutant [13]. To investigate the epigenetic component of the identified QTL region, Arabidopsis accessions displaying contrasting methylation patterns in the candidate region—*Ak-1*, *Appt-1*, *Fr-2*, *Gy-0*, *Cvi-0* and *Col-0*—were used.

The seeds of all materials were propagated simultaneously for at least one generation to minimise any environmental effects on seed quality due to the growth conditions of earlier generations.

## 2.2. Plant Culture of Selected Mutant Lines and Accessions

The root growth of selected lines and accessions (2.1.) was determined in vitro using vertical square petri dishes (120 × 120 × 17 mm) with six plants per plate and three plates per accession. The substrate was based on Estelle and Somerville [14], and mineral solutions were supplemented with 8% agar and 1% sucrose. The control treatment (9 mM nitrate) contained 5 mM $KNO_3$ and 2 mM $Ca(NO_3)_2$, while the low-nitrogen medium (0.4 mM nitrate) contained 0.2 mM $Ca(NO_3)_2$ and 5 mM KCl. Seeds were sterilised for 1 min in 70% ethanol followed by 1 min in 2% NaClO and four washing steps in sterile distilled water. Six seeds were regularly spaced 1 cm from the upper border of the dish. The seeds were stratified for 24 h at 4 °C, then transferred to a growth cabinet set to 16 h light (100 µmol $m^{-2}$ $s^{-1}$)/8 h dark, 20/18 °C for 14 days.

For the soil growth experiments of the selected lines and accessions with differing methylation patterns, a mixture of vermiculite (70%, from Kakteen Schwarz, Nürnberg, Germany) and growth substrate 1 (30%, from Klasmann-Deilmann GmbH, Geeste, Germany) was used and supplemented with Estelle and Somerville solution [14] by watering twice per week. Watering the pots with +N (9 mM nitrate) and −N (0.4 mM nitrate) solution resulted in 19.9 mg and 0.98 mg nitrate/100 g soil, respectively. Ammonium was in all cases tested below 0.13 mg/100 g soil (Agrolab Agrarzentrum GmbH, Leinefelde-Worbis, Germany). Plants were grown in climate-controlled growth chambers with a 16 h photoperiod at 20/18 °C and 60%/75% humidity for up to five weeks.

## 2.3. Phenotyping

Shoot biomass data for the GWAS of all accessions and mRILs were taken from [8], where the plants had been screened for biomass production in an agar plate-based, high-throughput procedure with a limited supply of nitrogen (0.05 mM $KNO_3$).

Root growth parameters were initially measured by a ruler and balance from plants grown on vertical agar plates. Six replicates per line were analysed per experiment and three independent experiments were performed.

At harvest, soil was removed from the plants grown in pots, and root length, total root weight, total rosette weight and total (floral) shoot weight were determined manually. Plant material was frozen in liquid nitrogen immediately afterwards for further analyses. Six replicates per line were analysed per experiment and three independent experiments were performed.

## 2.4. Genotyping and Sequencing

Genomic DNA was isolated using the DNeasy Plant Mini Kit (Qiagen, Hilden, Germany) from rosette leaves of 20-day-old plants. The analysis of the 102 accessions using 140 SNP markers [15] revealed that 38 of the accessions used in the experiments did not have 250K SNP data publicly available. DNA of the missing accessions was hybridised to the Affymetrix 250K SNP Array (performed by DNAVision, Charleroi, Belgium), and the raw data were subjected to the analysis pipeline established by Nordborg and colleagues [16] to ensure compatibility between the datasets. Version 75 was used for analyses [17]. The mixed RILs were genotyped using a custom 384plex Illumina Golden Gate Array (Table S3). Three additional SNP markers located in the candidate region on Chromosome 1 were designed using Primer3web version 4.1.0 (http://primer3.ut.ee/) [18] and analysed using SNaPShot technology (Applied Biosystems, Foster City, CA, USA) (Table S4).

For the detailed sequence analysis of the selected accessions, genomic DNA was extracted from the leaf and root tissue of soil-grown plants using the DNeasy Plant Mini Kit (Qiagen, Hilden, Germany) as described in the manufacturer's protocol. PCR was performed using the Taq DNA Polymerase kit (Qiagen, Hilden, Germany) in a Mastercycler gradient (Eppendorf, Hamburg, Germany), with 5 min at 95 °C; then 35 cycles of 30 s at 95 °C, 30 s at 60 °C, and 2 min at 72 °C; then a final elongation for 5 min

at 72 °C. After purification (Qiaquick, PCR purification kit, Qiagen, Hilden Germany), PCR fragments were Sanger sequenced at LGC Genomics GmbH (Berlin, Germany).

## 2.5. RNA Extraction, cDNA Synthesis and RT-qPCR

Total RNA was isolated from plants (28 DAS and 5 weeks of age) grown under sterile conditions, on solid agar substrate and from soil-grown plants. For each line, root and shoot samples were taken from plants grown under +N and −N conditions. Eight biological replicates were used per organ and treatment. Total RNA was isolated from 100 mg plant material using the RNeasy Plant Mini Kit (Qiagen, Hilden, Germany) for leaf samples and the Spectrum Plant Total RNA kit (Sigma-Aldrich, Steinheim, Germany) for root samples, following the manufacturer's protocols. RNA was dissolved in 30 μL DEPC-treated water and incubated with DNaseI (Roche, Mannheim, Germany). Total RNA concentrations were quantified using a Nanodrop® ND-1000 spectrophotometer (NanoDrop Technologies Inc., Wilmington, DE, USA). First-strand cDNA was synthesised by reverse transcription from the total RNA using the first strand RevertAid H Minus First strand cDNA synthesis kit (Fermentas, Vilnius, Lithuania).

Quantitative real-time measurements were performed using the POWER SYBR Green PCR Master Mix reagent (Applied Biosystems, Foster City, CA, USA) in an HT 7900 (Applied Biosystems, Foster City, CA, USA), according to the manufacturer's instructions. For each condition, three technical replicates and three to eight biological replicates were used. Transcript levels were determined by quantitative RT-PCR, and the raw threshold cycle values ($C_T$) for all samples were normalised against the $C_T$ values obtained for the reference transcript of the *ACTIN11* gene using the *qbase* software (Biogazelle). The primers used in this work were designed with the QuantPrime tool [19] and are listed in the supplementary data (Table S5).

## 2.6. Statistical Analyses

Statistical analyses were performed using GenSTAT 17th Edition (VSNi, Hempstead, UK). The adjusted means of the phenotypic data were estimated by a 2-factor ANOVA with "accession" and "nitrogen" as the main factors (accession + nitrogen + accession.nitrogen), and "experiment" as the blocking factor. The adjusted means of the qPCR expression data were estimated by a simple ANOVA with "genotype" as the main factor and "biological replicate" as the blocking factor. Significant differences were determined after Tukey multiple testing correction at 5%.

A GWAS was performed with a minor allele frequency (maf)>20, using ECMLM [20] as implemented in GAPIT [21,22]. The kinship matrix was chosen based on the QQ-plots obtained: the Loiselle kinship matrix was best suited to the accessions, and VanRaden was used for the mRILs. Running the module "Optimization for number of Principal Components (PCs) based on BIC" resulted in no PCs being included for the accessions, while 4 PCs had to be included for the highly structured mRILs.

Association studies in Arabidopsis usually employ 180 to 400 lines, although successful GWAS with population sizes of around 100 have been reported [23,24]. We used several criteria to select meaningful associations: $p < 0.0001$, the detection of several significant SNPs ("skyscraper") in the candidate genes [25], and validation in the independent mRIL population (low marker number, $p < 0.01$).

## 3. Results

### 3.1. Genome-Wide Association Mapping

The 102 accessions analysed previously [8] originate from different geographic regions (Figure 1). The plant biomass accumulation of accessions and mRILs grown under low N in agar has been shown to vary considerably [8]. Data for rosette biomass (shoot fresh weight) under low N were taken from [8] and used for statistical evaluation. To determine the genetic factors influencing the shoot fresh weight

under low N, a GWA analysis was performed separately for accessions and mRILs, using 106,402 and 536 SNP markers, respectively. A QTL for rosette biomass accumulation under low nitrogen on agar substrate was identified on Chromosome 1 (Figure 2A). A "skyscraper" delimited by the markers M1_20697610 and M1_20700224 (the marker name indicates the chromosome and position) contains eight significant ($p < 0.01$) marker-trait associations (MTAs), with the best association found for marker M1_20699404 ($p = 0.000033$) in the 3'-half of the coding region of *AT1G55430* (1:20699687–20697392). The confidence interval of ±5 kb flanking the most significant marker [26] included two adjacent genes annotated as "Cysteine/Histidine-rich C1 domain family protein" and oriented in tandem on the Crick strand. A third gene, *AT1G55420*, belonging to the same gene family and having the same orientation is located upstream of *AT1G55430*. For the 123 mRILs, no markers were initially present in the candidate region on chromosome 1. Therefore, three additional SNP markers localised within the candidate region were scored in the mRILs. These three markers: M1_20699406 ($p = 0.0337$), M1_20699440 ($p = 0.0059$) and M1_20699455 ($p = 0.0037$) rank among the most significant MTAs in the mRILs (Figure 2B).

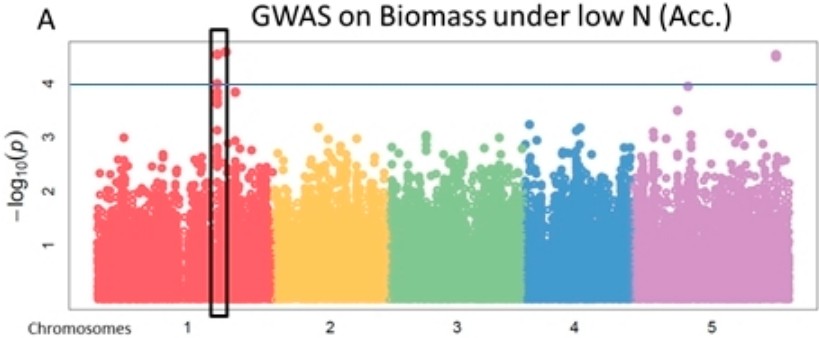

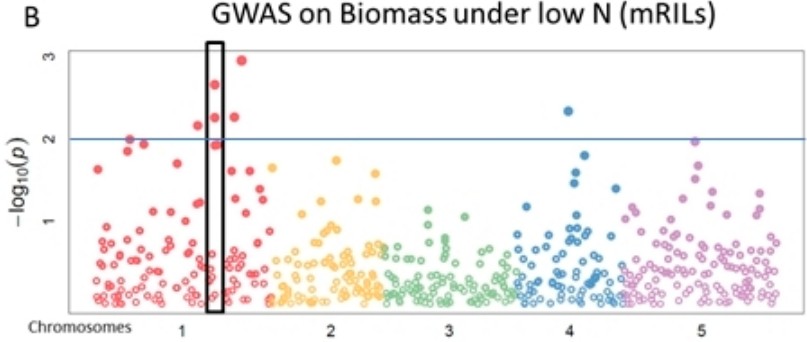

**Figure 2.** Identification of QTL for nitrogen use efficiency. (**A**) Manhattan blot of the genome-wide association study (GWAS) of rosette fresh weight under low nitrogen medium in 102 accessions using 100,274 SNP markers in GAPIT. Different Arabidopsis chromosomes are displayed in different colours. The box indicates the cluster of significant marker-trait associations (MTAs) on chromosome 1 (red) spanning two linked candidate genes. The horizontal line indicates the significance threshold ($p < 0.0001$). (**B**) Manhattan blot of genome-wide association study (GWAS) of rosette fresh weight on low nitrogen substrate in 123 mixed recombinant inbred lines (RILs) using 536 SNP markers in GAPIT. The horizontal line indicates the significance threshold ($p < 0.01$).

### 3.2. The Identified Region Contains Sequences Encoding Zinc-Finger Transcription Factors

To analyse the identified region in the context of the accumulation of biomass under low N on the molecular level, the region including the identified SNPs was inspected on the ComparativeGenomics platform [27] to reveal structural features, such as DNA methylation patterns, potential non-coding

RNAs and the genic structure of the identified genes and intergenic regions for the widely used reference accession *Col-0*. As depicted in Figure 3A, the region encodes three members of the zinc-binding cysteine/histidine-rich C1 domain family. *AT1G55420* was described as *EDA11* [28]. While *EDA11* contains an additional DMP1 domain (dentin matrix protein 1) at the N-terminus, all three proteins contain four cysteine-rich C1 domains, each proposed to bind two zinc ions (Figure S1). Based on the presence of the C1 domains binding by complexed zinc to DNA, these proteins are considered to bind diacylglycerol (DAG). The functional deletion of *EDA11* causes an abnormal number of nuclei and an arrest of the embryo sac's development [28]. All three genes consist of only one exon and share high homology (similarity >69.2% and <76.1% and identity >59.2% and <67.3%) (Figure 3A and Figure S1).

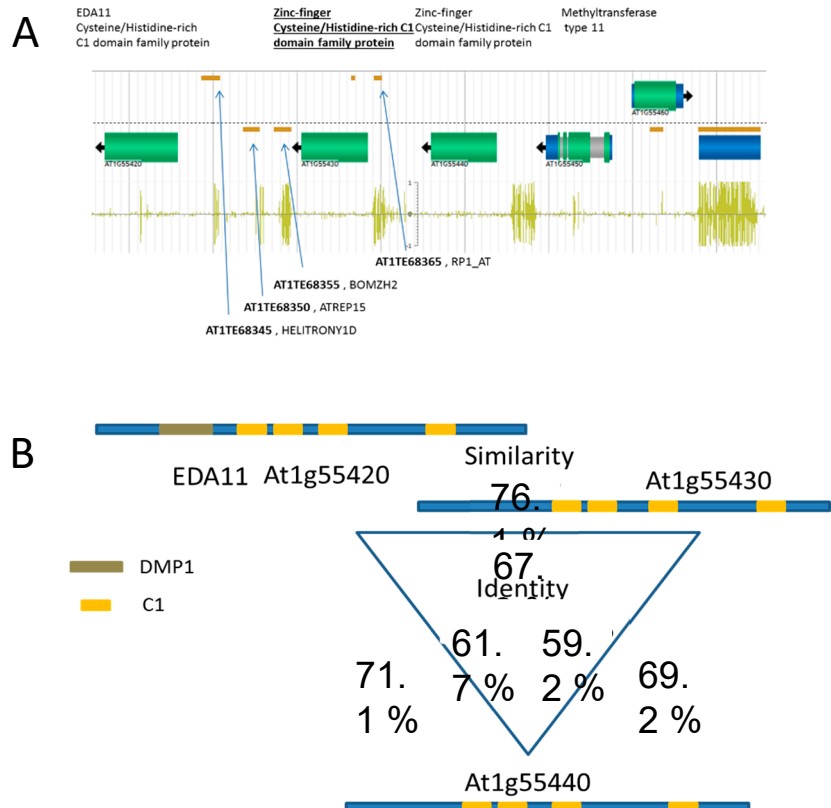

**Figure 3.** Region on chromosome 1 associated with the N use efficiency (NUE)-related growth trait QTL. (**A**) Schematic overview of the identified region encoding the genes *AT1G55420* (*EDA11*), *AT1G55430* and *AT1G55440*. Green: coding regions, arrows: orientations of the genes, grey: introns, blue: untranslated transcribed regions, orange: loci with similarity to repetitive structures. The lower line indicates DNA methylation, analysed by whole genome bisulfite sequencing in the reference accession *Col-0*. Data derived from GoGe Genome viewer. (**B**) Comparison of domain structure, similarity and identity of proteins encoded by *AT1G55420* (*EDA11*), *AT1G55430* and *AT1G55440*. DMP1: similar to dentin matrix protein 1, C1: cysteine-rich C1 domain with affinity to DAG (diacylglycerol). Identity between the genes is indicated (%) inside the triangle, with similarity outside.

The observation that RDR2 may be involved in the accumulation of biomass under low N was proposed earlier [8], and gives a first hint that RNA-directed DNA methylation [11,29] might be involved in the regulation of the identified region. Therefore, the DNA methylation patterns were inspected using the ComparativeGenomics database [30]. Repetitive DNA was detected in fragments of the intergenic regions: this repetitive sequence originates from transposable elements belonging to the helitron family. These regions are associated with high levels of DNA methylation (Figure 3A) and are often targeted by small heterochromatic RNA [11]. As the DNA methylation is overlapping

with the promotor sequences of the identified genes, this pattern makes them a target for regulation by transcriptional gene silencing [31].

For our further study, we used the available information on DNA methylation in the respective accessions [32] deposited in the ComparativeGenomics Database [30]. Here, we found that specific DNA methylation patterns occurred in this region (Figure S2). The majority (52%) of inspected accessions (N = 137) contained methylated promoter regions in all three candidate genes. To investigate the effect of the methylation pattern on biomass production under low N, six accessions contrasting in their DNA methylation patterns were chosen. *Col-0* and *Cvi-0* were selected to represent the majority of ecotypes with substantial DNA methylation in the promoter region of the three zinc-finger genes. *Appt-1* and *Gy-0* were selected because of their absence of detectable DNA methylation (Figure 4). *Ak-1* and *Fr-2* are accessions with DNA methylation in the coding regions of the genes. While DNA methylation in the promoter region can cause transcriptional gene silencing, methylation in the gene body is supposed to be a "footprint" of post-transcriptional gene silencing [33].

According to data displayed (eFP browser; [34]), the expression of all three genes is generally low. *AT1G55420* and *AT1G55430* show the highest expression in roots, while *AT1G55440* displays the highest expression values in dry seeds but also in seedling roots. The eFP Browser also indicates natural variation in the expression levels of all three genes. *AT1G55430* is expressed in the root elongation zone (eFP, [35]), but does not respond to high nitrate [36], while the expression of *AT1G55420* and *AT1G55440* increases in lateral root caps under high nitrate supply (eFP, [36]).

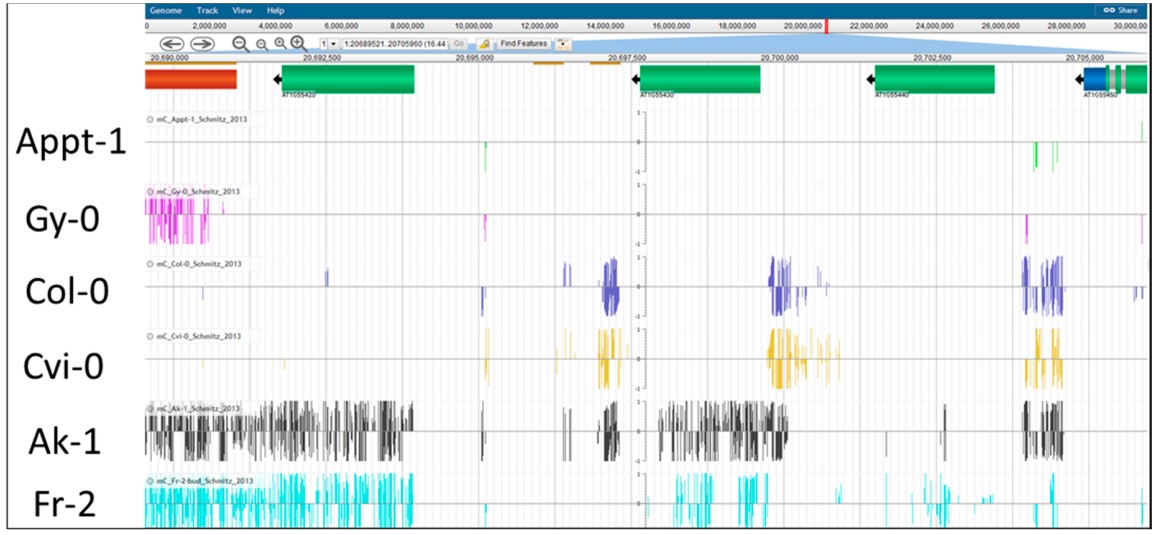

**Figure 4.** DNA methylation in the QTL region. Appt-1 and Gy-0 represent accessions without detectable cytosine methylation in the promoter regions of *AT1G55420*, *AT1G55430* and *AT1G55440*. Col-0 and *Cvi-0* represent the majority of accessions with a typical DNA methylation pattern in the promoter regions of the respective genes. *Ak-1* and *Fr-2* are special accessions with gene body specific DNA methylation detected.

### 3.3. Plant Growth of Selected Accessions under Nitrogen Limiting Conditions

After the identification of the aforementioned QTL, the influence of the DNA methylation pattern was tested. Therefore, accessions with different methylation patterns were selected and analysed in more detail using a novel experimental setup. In this experimental setup, plants were grown under control and nitrogen-deficient soil conditions in a mixture of 70% vermiculite and 30% soil substrate to minimise the initial amount of nitrate and ammonium. The available nitrogen was supplemented by watering the plants with a nutrient solution, resulting in amounts of 19.9 or 0.98 mg nitrate/100 g soil for +N or −N treatments, respectively (Figure S3). This approach had the advantage that (i) plants could be grown till their reproductive phase and (ii) the applied conditions were close to those in soils.

The phenotypes and growth performance of the plants were monitored. Representative pictures of accessions *Col-0*, *Cvi-0*, *Ak-1*, *Fr-2*, *Appt-1* and *Gy-0* 44-days after sowing (DAS) are shown in Figure 5. Under nitrogen limitation (−N), the plants displayed smaller total leaf areas. As described earlier [8], superior plants were defined as those accumulating the most biomass under nitrogen limitation.

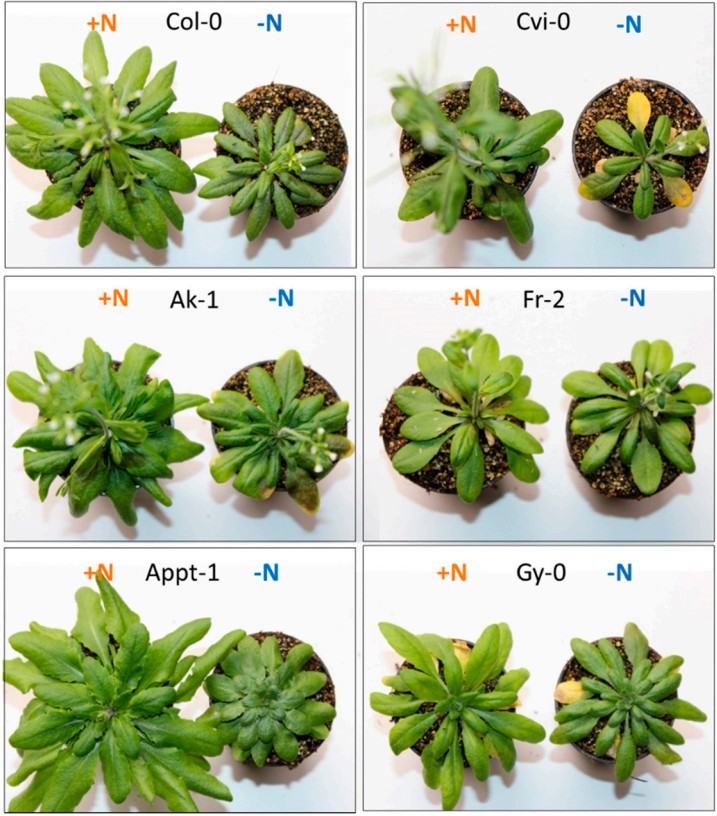

**Figure 5.** Representative pictures of plants grown under +N (left/orange) and −N (right/blue) conditions in a vermiculite-based substrate. Shown are representative plants of the *Arabidopsis thaliana* accession *Col-0*, *Cvi-0*, *Ak-1*, *Fr-2*, *Appt-1* and *Gy-0* grown for 44 days on 70% vermiculite/ 30% soil mixture supplemented with Estelle and Somerville medium to achieve +N (9 mM nitrate) and −N (0.4 mM nitrate) conditions.

Rosette biomass (fresh weight: FW, Figure 6A) and the root length of the main root (root length: RL, Figure 6B) were estimated 44 days after sowing at the end of the experiment, using 2-factor ANOVA. We found highly significant differences ($p < 0.001$) for all three factors (accession, nitrogen and the interaction term accession.nitrogen) for fresh weight, indicating differential growth reactions of the accessions to limited nitrogen (Table S6 and Figure 6A). Based on the biomass data, *Appt-1* and *Gy-0* were considered good performers and *Col-0* and *Cvi-0*, bad performers. As a general response to nitrogen deficiency, Arabidopsis plants have been described to develop longer roots [37]. We found highly significant differences ($p < 0.001$) according to accession and the interaction term accession.nitrogen for root length (Table S6 and Figure 6B), but only *Col-0* and *Cvi-0* displayed longer roots under low N than under sufficient N (Figure 6B). Furthermore, it should be noted that the purple or yellow colours of the older leaves in the stress-sensitive lines (*Col-0* and *Cvi-0*), especially under low N, is indicative of N deficiency (Figure 5).

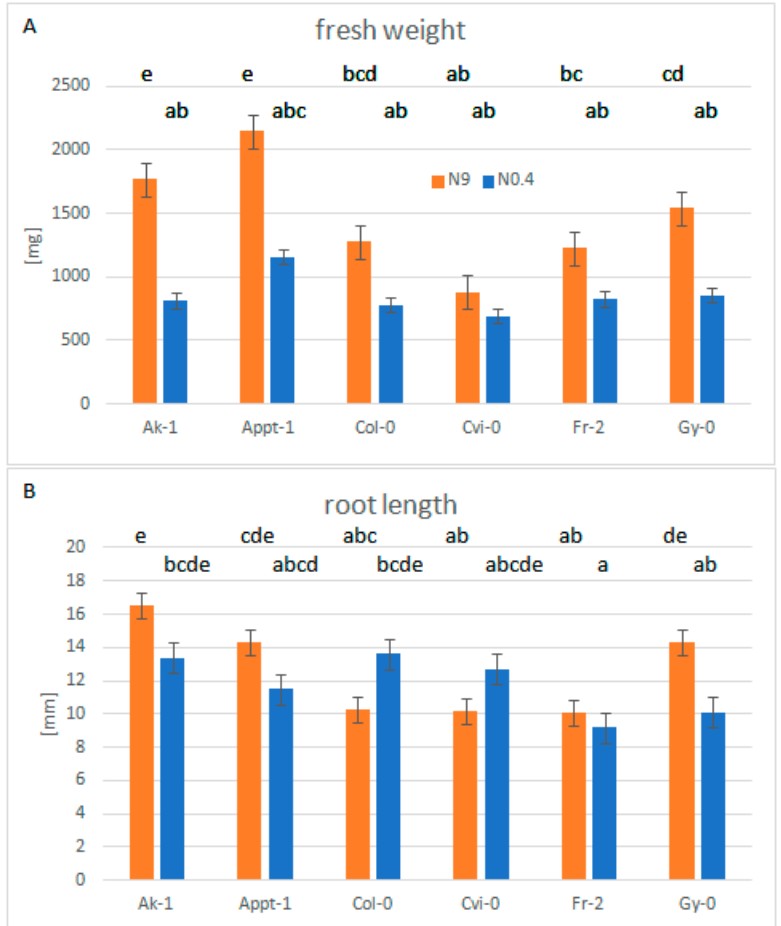

**Figure 6.** Rosette fresh weight and root length of 44-day old plants as parameters for plant performance under nitrogen deficient conditions. (**A**) Rosette fresh weight (mg) of selected accessions grown under +N (orange) and −N (blue) conditions in a vermiculite-based substrate. (**B**) Root length (mm) of selected accessions grown under +N (orange) and −N (blue) conditions in vermiculite-based substrate. Bars show adjusted means and standard errors estimated over three experiments using a 2-factor ANOVA (*n* = 6). Different letters indicate significant differences (*p* < 0.05 after Tukey correction).

*3.4. Expression Analysis of the Candidate Genes under Nitrogen Limiting Conditions*

As the investigated QTL encompasses three genes, the transcriptional response of *AT1G55420*, *AT1G55430* and *AT1G55440* was quantified by detecting the relative mRNA abundance using RT-real-time qPCR. The mRNA abundance of the identified genes was analysed in leaves and roots, and found to be root-specific. Therefore, samples were taken from the root tissue of soil-grown plants cultivated under +N and −N conditions. While all three genes were expressed under +N conditions, the expression was increased under nitrogen deficiency. The rate of induction varied from 1.5 to 6-fold (*AT1G55420,* Figure 7A), 1.5 to 7-fold (*AT1G55430*, Figure 7B) and 1 to 8-fold (*AT1G55440,* Figure 7C). In the accessions with substantial DNA methylation in the promoter region (*Col-0, Cvi-0* and *Ak-1*), the induction was strongest. The accessions without detectable DNA methylation showed almost no or only minor induction in response to nitrogen deficiency in the tested root tissue. In *Fr-2*, the only accession with gene body methylation, an induction of *AT1G55420* and *AT1G554440* was detectable, while *AT1G55430* was almost not expressed or inducible.

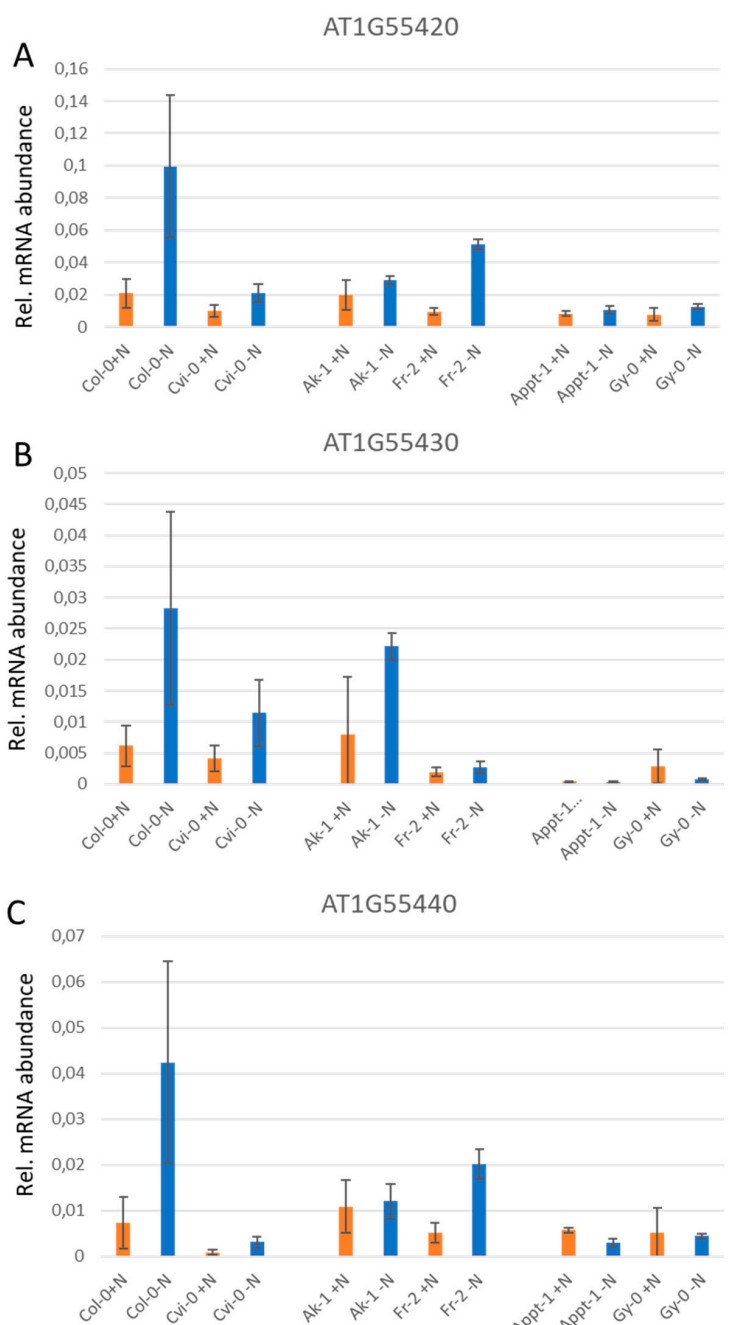

**Figure 7.** Relative mRNA abundance of identified genes in the QTL region under +N and -N conditions. Analysed were selected accessions with differences in the methylation pattern. Left part: detected methylation signatures, right part: absent methylation signature. Bars indicate the means of 6 biological replicates (3 technical replicates) with standard deviations. (**A**) mRNA abundance of *AT1G55420* (*EDA11*), determined by real-time qPCR, relative to *ACTIN11* mRNA in root tissue. (**B**) mRNA abundance of *AT1G55430*, determined by real-time qPCR, relative to *ACTIN11* mRNA in root tissue. (**C**) mRNA abundance of *AT1G55440*, determined by real-time qPCR, relative to *ACTIN11* mRNA in root tissue.

*3.5. Analysis of Mutants Deregulated for Their DNA Methylation Pattern in the QTL Region*

As the involvement of RDR2 was proposed earlier [8], a subset of T-DNA insertion mutants affecting DNA methylation patterns was selected to investigate the effect on biomass accumulation under low N. In order to investigate whether RDR2 is directly or indirectly (via heritable modification of DNA methylation) involved in the NUE-related response, *rdr2-1* mutant plants (*Col-0* background, [11])

differing in their methylation pattern for the QTL region were investigated for their performance under N-limiting conditions. The loss-of-function mutants *rdr2* and *suvh2*, and the T-DNA insertion mutants *rdr6* and *suvh9* were analysed. While *rdr2* and *suvh2* are hypomethylated at the QTL region (Supplementary Figure S4) in the asymmetric cytosine site (CHH) context, in *rdr6* and *suvh9*, no difference in the methylation pattern was observed. It should be noted that the difference in methylation is mainly restricted to the asymmetric cytosine sites (CHH), considered to be the result of RdDM.

As depicted in Figure 8, in *suvh2*, the gene expression of all three candidate genes was low and no induction under nitrogen deficiency was detectable. The analysis of plants with impaired an RdDM pathway—in particular, with hypomethylation of the QTL region (Supplementary Figure S4—indicates interplay between DNA methylation and *AT1G55430* inducibility. As depicted in Figure 8, *rdr2*, *suvh2* and *suvh2/9* double mutants are not capable of inducing *AT1G55430* under nitrogen deficiency.

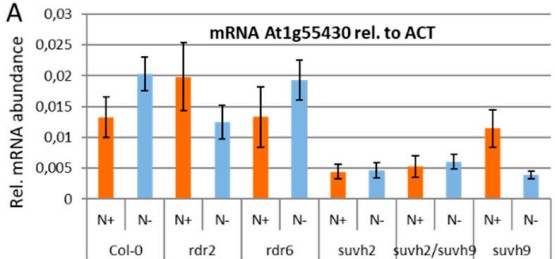

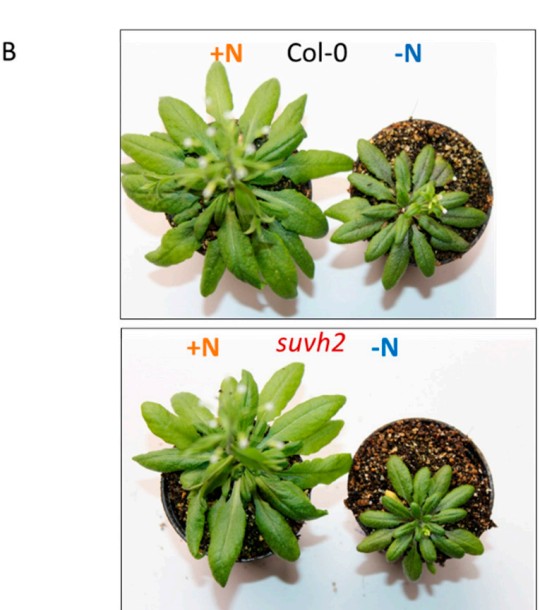

**Figure 8.** Analysis of selected mutants for RNA directed RNA polymerase (RDR) function and histone methyltransferase (SUVH) function under limited N conditions. (**A**) mRNA abundance of *AT1G55430* in selected mutants for RDR function and SUVH function on +N (orange) and −N (blue), determined by real-time qPCR, relative to *ACTIN11* mRNA in root tissue. Bars indicate the means of 6 biological replicates (3 technical replicates) with standard deviations. (**B**) Representative pictures of plants grown under +N (left/orange) and −N (right/blue) conditions in a vermiculite-based substrate. Shown are representative plants of the *Arabidopsis thaliana* accession *Col-0*, and the T-DNA insertion mutant *suvh2* grown for 44 days on a 70% vermiculite/30% soil mixture supplemented with Estelle and Somerville solution to achieve +N (9 mM nitrate) and −N (0.4 mM nitrate) conditions.

As the hypomethylation of *Appt-1* and *Gy-0* correlated with low expression levels and almost no inducibility under nitrogen deficiency, the promoter region and 5' region of *AT1G55430* were amplified by PCR and sequenced, resulting in the detection of three deletions in the sequence of *Appt-1* (Supplementary Figure S5). These deletions may cause the lack of transcriptional inducibility of *AT1G55430* under nitrogen deficiency.

## 4. Discussion

The aim of our study was the identification of a genomic region in Arabidopsis contributing to superior plant performance under low nitrogen conditions in soil. In order to reveal the molecular background of the identified QTL, the corresponding genes should be analysed for their contribution to the N stress response. It should be pointed out, here, that biomass production is a complex trait that could be influenced by nitrogen uptake, root development and also nitrogen use efficiency.

### 4.1. Quantitative Trait Locus for Biomass Production under Nitrogen Limitation

In order to identify novel genes that are involved in plant performance under low nitrogen conditions, 102 *Arabidopsis thaliana* accessions were previously analysed for their biomass production in an agar plate-based experimental setup [8].

Our genome-wide association study revealed a region on Chromosome 1 potentially involved in a regulatory function for biomass accumulation and plant growth performance under low nitrogen. Within the identified region, three genes were detected. All three genes encode zinc-finger proteins. *AT1G55420* is described as *EDA11* [28], with a functional connection to embryonal plant development. To date, no molecular function has been assigned to these three genes. Zinc-finger proteins can be involved in various processes: they are required for DNA recognition, RNA packaging, transcriptional activation, protein folding and assembly, and lipid binding [38]. As transcriptional regulators, they are able to induce [39] or suppress their target genes. In *Arabidopsis thaliana*, 176 members of the Cys2His2-type zinc-finger protein family have been identified [39]. Seventy-three of these genes encode C1 domain proteins and are classified into nine subclasses. The three genes from the identified gene cluster are grouped in subfamily five [40]. They are supposed to be involved in the integration of abiotic stress-mediated signaling pathways. All three identified genes encode a C1 domain that can bind to diacylglycerol (DAG). DAG is the precursor of triacylglycerol (TAG) and also functions as secondary messenger [41]. TAG is considered a storage lipid compound that accumulates under nitrogen starvation [42–44]. From an economic perspective, TAG is the predominant component of the seed oil of oleaginous plants and extensively studied for its application in biodiesel production [45]. As *Brassica napus* is one of the main crops used for biodiesel production, the identified region might harbour an interesting regulator involved in the accumulation of TAG under nitrogen-deficient conditions.

Arabidopsis accessions that perform poorly under nitrogen deficiency (e.g., *Col-0*) with respect to biomass accumulation show substantial expression of the candidate genes and strong induction under nitrogen limitation (Figure 7). This further supports the idea that the identified genes in the detected QTL are involved in the nitrogen stress response in *Arabidopsis thaliana*.

*AT1G55430* was detected to be expressed in the roots of *Col-0*, *Cvi-0* and *Ak-1* under nitrogen limitation (Figure 7B). In *Appt-1*, *Gy-0* and *Fr-2*, this transcriptional induction under N limitation was not detectable. In these accessions without transcriptional induction, the effect of generated biomass under nitrogen limitation (Figure 6A) and also anthocyanin production (Figure 5) was weaker. Whether the protein encoded by the gene identified here is part of a signaling cascade or directly involved in DAG/TAG production will be investigated in the future.

### 4.2. DNA Methylation in the Promotor Region Correlates with the Transcriptional Response of the Candidate Gene(s)

A general statement about the effect of the promoter methylation of the genes located in the identified QTL on the nitrogen stress response is difficult to make. As the plant biomass contributes

to the plant's performance under nitrogen deficiency, the detectable stress effect also depends on the biomass of the accession under consideration. As the T-DNA insertion lines affecting DNA methylation pathways also differ by plant biomass production under +N control conditions (*rdr2*), the respective nitrogen use under –N conditions will vary, and the corresponding stress response will be reduced due to the smaller plant size. Furthermore, it was found that in some cases, where no DNA methylation was noted (*Appt-1*), no corresponding dataset was available. This is caused by the applied method: as only reads of methylated regions result in a perfect mapping after bisulfite conversion, only these are reliably mapped. It should be noted, here, that regions without mapped reads and unmethylated regions are displayed in the same way (Figure 4). As a consequence, the existence of the respective region has to be tested. As was found for *Appt-1,* a deletion in the promoter of *AT1G55430* resulted in the absence of methylation. This deletion caused the loss of gene expression and inducibility of *AT1G55430* under nitrogen limitation. As plants with missing expression of this gene are larger in biomass, it was concluded that *AT1G55430* makes a major contribution within this gene family, with a suppressive function on plant biomass production under nitrogen-limiting conditions.

Beside all these limitations, an influence of DNA methylation on the transcription level was found. Although the detected pattern of promoter methylation strongly indicated a regulation via transcriptional gene silencing, substantial transcription was only detected in the presence of methylation. Therefore, the detected pattern of methylation and correlating gene expression suggested the requirement of DNA methylation for proper gene induction. In particular, a change in RdDM-mediated asymmetric cytosine methylation appears to affect the transcription. Transcriptional activation that requires DNA methylation and the involvement of SUVH proteins has been described recently [46]. An additional strong indication for the involvement of RNA-directed DNA methylation was the identification of RDR2 [11] as a gene involved in NUE [8]. Therefore, these mutants affecting the RdDM pathway were investigated. According to the WGBS data, methylation in the asymmetric sequence context in the QTL is not present in *suvh2* [13] and *rdr2*. The analysis of the *suvh2* (and *suvh2/9* double mutant) loss-of-function mutant revealed that the plant's performance in the absence of methylation and reduced expression of the candidate genes was superior concerning biomass accumulation. This finding further supported the idea of the requirement of DNA methylation for gene transcription in this region.

### 4.3. Transfer of Gained Knowledge to Other Crop Plants

Our molecular analyses revealed genotype-specific differences in the expression of a, so far, poorly characterized gene that correlates with NUE and is involved in epigenetic regulation. The identified gene family encoding zing-finger proteins is described here for the first time as being related to the plants' response to low nitrogen stress. The molecular analysis of this gene family in other (crop) species might lead to a new component of knowledge regarding low nitrogen responses.

Based on the knowledge gained here and the results obtained in previous work [8], we found that the expression of genes involved in TAG biosynthesis is influenced by nitrogen availability (Supplementary Figure S6). The genes involved in DAG synthesis (*PAH1* and *PAH2*) show a stronger induction in the stress-responsive lines. Additionally, the reduced presence of *SDP1* mRNA, encoding an enzyme mediating the reverse reaction (TAG to DAG), indicates a stronger accumulation of TAGs under nitrogen deficiency as a consequence of the nitrogen stress response. Furthermore, also in this dataset, the *AT1G55430* expression pattern could clearly distinguish inferior and superior lines for biomass production (Figures 6 and 7).

As the accumulation of TAGs is of utmost importance for biofuel production with *Brassica napus*, the identified gene in the QTL represents a promising regulatory hub, connecting DAG sensing and TAG production with the nitrogen stress response.

**Supplementary Materials:** The following are available online at http://www.mdpi.com/2073-4395/10/5/636/s1, Figure S1: Alignment of *EDA11*, *AT1G55430* and *AT1G55440* amino acid sequences. Figure S2: Patterns of DNA methylation in the QTL region. Figure S3: Nitrogen content in vermiculite substrate. Figure S4: Methylation pattern in *suvh2/9* and *rdr2/6* mutant plants. Figure S5: Promotor sequence of *Appt-1 AT1G55430*. Figure S6: Gene expression related to triacylglycerol synthesis. The transcription of genes involved in synthesis of DAG/TAG under nitrogen deficiency. Table S1: Information about accessions used in this study. Table S2: Information about crosses prepared for mixed Recombinant Inbred Lines (mRILs). Table S3: Information about markers used in GWAS for mRILs. Table S4: Primers and location of SNPs for SNaPShot technology. Table S5: Primers used in this study. Table S6: 2-factor ANOVA of fresh weight and root length under different N regimes.

**Author Contributions:** Conceptualization, M.K., R.C.M., N.v.W. and T.A.; formal analysis, M.K., R.C.M., D.K., L.-M.K., Z.J.; writing—original draft preparation, M.K.; writing—review and editing R.C.M., N.v.W. and T.A. All authors have read and agreed to the published version of the manuscript.

**Funding:** This work was funded by the GABI-FUTURE grant 0315064 of the Bundesministerium für Bildung und Forschung (Federal Ministry of Education and Research (BMBF), Germany) to TA.

**Acknowledgments:** We thank Andrea Apelt, Iris Fischer, Monika Gottowik, Marion Michaelis, Sabine Struckmeyer and Gunda Wehrstedt for excellent technical support.

**Conflicts of Interest:** The authors declare no conflict of interest.

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
