# Peer review of "Epigenetic Variation at a Genomic Locus Affecting Biomass Accumulation under Low Nitrogen in Arabidopsis thaliana"

_agronomy, doi:10.3390/agronomy10050636_

Round 1

Reviewer 1 Report

Dear authors,

The answers given and the modifications made in the paper are satisfactory.

There all still some typos to correct in some places and the text must carefully checked again : inadequate capital letter, missing bracket, the threshold lines are indicated "vertical" instead of "horizontal" in the legend of figure 2, the anova table in supplementary material deserves a better layout...

Appart from this format details, I have no other comment.

Best regards

Author Response

Gatersleben, 21.4.2020

Dear editor, dear reviewer,                                                              

On behalf of the authors, I have to say that we are grateful for your encouraging assessment of our manuscript and would like to especially thank the anonymous reviewer for their critical evaluation and very helpful suggestions for improvement. We have carefully edited our manuscript according to the recommendations made by the reviewer and hope that it is now suitable for final acceptance by agronomy.

Please find below a listing of the comments made by the reviewers and descriptions of how we dealt with them when revising the manuscript.

Dear authors,

The answers given and the modifications made in the paper are satisfactory.

  We again thank the anonymous reviewer for the encouraging assessment of our manuscript and constructive criticism.

There all still some typos to correct in some places and the text must carefully checked again : inadequate capital letter, missing bracket, the threshold lines are indicated "vertical" instead of "horizontal" in the legend of figure 2, the anova table in supplementary material deserves a better layout...

  We carefully corrected the manuscript, corrected the above indicated mistakes, changed the legend of figure 2 reformatted the supplementary table 2.

Appart from this format details, I have no other comment.

Best regards

  We again thank the editor and the reviewer for their helpful suggestions and wish you good health in this times.

Best regards,

Markus Kuhlmann

Reviewer 2 Report

The authors have addressed all my previous comments.

Author Response

Gatersleben, 21.4.2020

Dear editor, dear reviewer,                                                              

On behalf of the authors, I have to say that we are grateful for your encouraging assessment of our manuscript and would like to especially thank the anonymous reviewer for their critical evaluation and very helpful suggestions for improvement. We have carefully edited our manuscript according to the recommendations made by the reviewer and hope that it is now suitable for final acceptance by agronomy.

Please find below a listing of the comments made by reviewer1 and descriptions of how we dealt with them when revising the manuscript.

Dear authors,

The answers given and the modifications made in the paper are satisfactory.

We again thank the anonymous reviewer for the encouraging assessment of our manuscript and constructive criticism.

There all still some typos to correct in some places and the text must carefully checked again : inadequate capital letter, missing bracket, the threshold lines are indicated "vertical" instead of "horizontal" in the legend of figure 2, the anova table in supplementary material deserves a better layout...

We carefully corrected the manuscript, corrected the above indicated mistakes, changed the legend of figure 2 reformatted the supplementary table 2.

Appart from this format details, I have no other comment.

Best regards

We again thank the editor and the reviewer for their helpful suggestions and wish you good health in this times.

Best regards,

Markus Kuhlmann

This manuscript is a resubmission of an earlier submission. The following is a list of the peer review reports and author responses from that submission.

Round 1

Reviewer 1 Report

Dear author

The presented paper is well written and have many scientific qualities. Nevertheless, I have some recommendations and questions concerning both the format and the scientific background.

Format

  • I would have appreciated not to find a beginning of conclusion at the end of the introduction (lines 71, 72 and 73).
  • I think the paper needs a “wrap-up” conclusion section after the discussion
  • Some typos exist and must be corrected (for example, I’m not sure that the reference to the figure 5 is good in line 310).

Scientific background

  • It is not clear how many plants have been soil tested in pots : the 2 whole panels (ie the same ones that have been tested in Agar) or a subset? Unless I am mistaken, this info is not delivered (or only available in the reference [7]). Anyway, if it does exist in the paper, it must be displayed more clearly for the reader.
  • If the whole panels have also been tested in pots, you must explain why the GWAS have not been performed for biomass measured in pots too.
  • Concerning the GWAS: 1) please give some arguments to justify that your number of accessions (102 or 123) is statistically high enough to perform a good analysis / 2) please indicate with an horizontal line in figures 2 A and 2 B the significance -log10(P) threshold you used and how it has been calculated / 3) did you try to perform a GWAS with a trait related to the tolerance to N deficiency (for example, a ratio biomass -N/biomass +N) ? If not, what is the reason you didn’t try? If yes, what was the result and why not use it in the paper ?
  • Concerning the pictures-based phenotyping, I understand you only used it to visually assess the effect of N deficiency on growth and leaf symptoms. Why don’t you try to quantify some new interesting traits like LAI by a picture analysis software?

 Best regards

Author Response

 Gatersleben, 9.4.2020                                                    

Dear Editor, dear reviewers, 

On behalf of the authors, I have to say that we are grateful for your encouraging assessment of our initial manuscript and would like to especially thank the anonymous reviewers for their critical evaluation and very helpful suggestions for improvement. We have carefully edited our manuscript according to the recommendations made by the reviewers and hope that it is now suitable for final acceptance by agronomy.

Please find below a listing of the comments made by the reviewers and descriptions of how we dealt with them when revising the manuscript.

Dear author

The presented paper is well written and have many scientific qualities. Nevertheless, I have some recommendations and questions concerning both the format and the scientific background.

We thank the anonymous reviewer for the encouraging assessment of our manuscript and constructive criticism. We hope that we addressed all raised points and could improve the manuscript.

Format

  • I would have appreciated not to find a beginning of conclusion at the end of the introduction (lines 71, 72 and 73).

Conclusions were removed from the introduction and added in the wrap up section in the discussion.

  • I think the paper needs a “wrap-up” conclusion section after the discussion

Subsuming section was added in the beginning of 4.3..

  • Some typos exist and must be corrected (for example, I’m not sure that the reference to the figure 5 is good in line 310).

Typos were corrected and reference in line 310 figure 5 was changed to Figure 8

Scientific background

  • It is not clear how many plants have been soil tested in pots : the 2 whole panels (ie the same ones that have been tested in Agar) or a subset? Unless I am mistaken, this info is not delivered (or only available in the reference [7]). Anyway, if it does exist in the paper, it must be displayed more clearly for the reader.

The section was partially rewritten and explanatory text was added:

L99 Shoot biomass data for GWAS of all accessions and mRILs was taken from [7], where the plants had been screened for biomass production in an agar plate-based high-throughput procedure with limited supply of nitrogen (0.05 mM KNO3)….

L111 For soil growth experiments of the selected lines and accessions with differing methylation patterns,…

  • If the whole panels have also been tested in pots, you must explain why the GWAS have not been performed for biomass measured in pots too.

Only the selected lines and accessions with differing methylation patterns were grown in the Vermiculite/soil mixture in this study. The previous study had shown that agar and soil substrates delivered very different results, and the majority of the deep phenotyping was (and had to be) done on agar substrate. Our goal here was to reproduce data from agar-grown plants with pot-grown plants to profit from the longer growth period of up to 5 weeks compared to 2 weeks.

  • Concerning the GWAS: 1) please give some arguments to justify that your number of accessions (102 or 123) is statistically high enough to perform a good analysis / 2) please indicate with an horizontal line in figures 2 A and 2 B the significance -log10(P) threshold you used and how it has been calculated / 3) did you try to perform a GWAS with a trait related to the tolerance to N deficiency (for example, a ratio biomass -N/biomass +N) ? If not, what is the reason you didn’t try? If yes, what was the result and why not use it in the paper ?

1) Text added (2.6): Association studies in Arabidopsis usually employ 180 to 400 lines, although successful GWAS with population sizes of around 100 have been reported [36, 37]. We used several criterions to select meaningful associations: p<0.0001, the detection of several significant SNPs (‘skyscraper’) in the candidate genes [38], and the validation in an independent population (mRILs with p<0.01).

2) Lines added to figure 2.

The p-value thresholds are based on experiences with GWAS using GenStat.

3) The accessions and mRILs were only screened on low N substrate, therefore performing GWAS on the ratio is unfortunately not possible.

Concerning the pictures-based phenotyping, I understand you only used it to visually assess the effect of N deficiency on growth and leaf symptoms. Why don’t you try to quantify some new interesting traits like LAI by a picture analysis software?

As the trait of color change was not analyzed in detail, we removed this section. The correlation of anthocyanin accumulation and leaf coloring as well as other leaf traits will be analyzed in detail in future experiments.

 Best regards

We again thank all reviewers for the constructive criticism. We hope that we could address all raised points and improve the manuscript for final acceptance in Agronomy.

Yours Sincerely,

Markus Kuhlmann

Reviewer 2 Report

General comments:

It is not clear enough weather authors selected Arabidopsis accessions with different methylation patters or because of superior response to conditions of limiting nitrogen (or both); and if this selection process was part of this study or previous work.

Plant total fresh weight is a complex trait results of many interactive processes, and its response under N limiting conditions may have nothing to do with nitrogen use efficiency, and more with canopy developmental traits or soil exploration. Quantifying the total N uptake and a measure of nitrogen use efficiency would be better indicated for the goals the authors proposed. Also the results are influenced by the light conditions during experiments. Can the authors elaborate on the limitations of their current approach?

Authors need to better justify in the introduction and discussion how the approach used and the findings from this study can advance development of crops with increased nitrogen use efficiency.

The authors emphasize discussion of plant color, which was not a trait measured or presented in the article. This should be avoided and authors should limit their conclusions and discussion to data obtained  in their study.

Specific comments:

Line 34 – 49. This introduction is too general. Please rewrite to justify the approach in the present paper, and not the need to improve NUE in general.

Line 50 – Replace “the biomass” with “biomass”

Line 70 – accessions were grown in either agar or soil, or under both conditions?

 Line 83 to 85 – Justify why these insertions were selected

Line 85 – 86 – not clear weather authors selected accessions with different methylation patters as part of the work in this study or previous work

Line 90 – What selected lines?

Line 95. Replace ‘12’ with ‘Twelve’.

Line 167-168 – Rewrite sentence for more clarity.

Line 107 – Please provide additional details about the shoot data collected and in what treatments. Was it collected only under the low N? the reference cited says: “analysis of shoot growth of the populations on low N soil was performed according to Tschoep et al. (2009)”. Right now readers need to go to an additional source to understand the methods.

Line 108 – replace “assessed’ by the precise measurements that were taken. How often, and in what treatments?

Line 110 – 111 – What type of data was obtained from imaging? This data is not presented in the article. Present the data or remove these protocols from the methods.

Line 155 – 157 – Describe what were the factors considered in the ANOVA and if they were considered fixed or random. How were the replicated experiments treated in the model?

Line 166 – Replace “plant performance” by other traits or processes that were affected in the study cited.

Line 167-168 – Rewrite sentence for more clarity.

Line 253 – Not clear if the accessions selected were selected because of being superior, or because of their different methylation patterns in the three candidate genes.

Line 254 – the authors seem to be presenting results from another paper. The text must be more clear indicating what is part of this paper, or from previous work.

Line 261 – throughout the article there is inconsistency in naming biomass, biomass mass, fresh weight, shoot mass,  ‘Biomass mass’ is incorrect. Use shoot fresh mass throughout the article, biomass usually refers to dry weight, and on an area basis.

Line 261 – What is maximal root extension? Root length? Use appropriate and specific names to define traits measured.

Line 267 – Replace   “decrease in shoot biomass of growth resulting in the smallest plant” by “decrease in shoot fresh mass”

Figure 6  and associated text in results section – Statistics from the ANOVA should be provided in the Figure, and mentioned in the presentation of results. Avoid presenting results and tendencies and focus on the statistical analysis (e.g. Line 267-268). Results from root data are not presented in the results section.

Line 361 – this the first mention that Col-0 performs poorly under N limiting conditions. Poorly relative to other accessions? Where is this data presented or discussed. There is not enough context given.

Line 326-327 – What is a reduced stress response?

Line 327-328 – the color differences are not apparent in Figure 8.

Line 367 – the color differences are not apparent in Figure 5. Why is data obtained from imaging not used to obtained quantitative results, instead of relying on a subjective interpretation of color?

Line 374-375 – This paper does not: 1) analyze how production of anthocyanin relates with biomass accumulation, and 2) measure in any way red color (or at least this data is not presented). Authors should not make any interpretation or conclusion based on color or anthocyanin accumulation, or if it is independent or not from biomass accumulation.

Author Response

Gatersleben, 9.4.2020

Dear Editor, dear reviewers, 

On behalf of the authors, I have to say that we are grateful for your encouraging assessment of our initial manuscript and would like to especially thank the anonymous reviewers for their critical evaluation and very helpful suggestions for improvement. We have carefully edited our manuscript according to the recommendations made by the reviewers and hope that it is now suitable for final acceptance by agronomy.

Please find below a listing of the comments made by the reviewers and descriptions of how we dealt with them when revising the manuscript.

Dear author

The presented paper is well written and have many scientific qualities. Nevertheless, I have some recommendations and questions concerning both the format and the scientific background.

We thank the anonymous reviewer for the encouraging assessment of our manuscript and constructive criticism. We hope that we addressed all raised points and could improve the manuscript.

Open Review

English language and style

( ) Extensive editing of English language and style required
( ) Moderate English changes required
(x) English language and style are fine/minor spell check required
( ) I don't feel qualified to judge about the English language and style

Comments and Suggestions for Authors

General comments:

It is not clear enough weather authors selected Arabidopsis accessions with different methylation patters or because of superior response to conditions of limiting nitrogen (or both); and if this selection process was part of this study or previous work.

  • The selection process followed the structure of the manuscript: We first used the entire set of accessions to perform the GWAS, identified the QTL region and selected the accessions based on their contrasting methylation patterns in order to test our hypothesis. An additional explanatory sentence was added in line 252ff.

Plant total fresh weight is a complex trait results of many interactive processes, and its response under N limiting conditions may have nothing to do with nitrogen use efficiency, and more with canopy developmental traits or soil exploration. Quantifying the total N uptake and a measure of nitrogen use efficiency would be better indicated for the goals the authors proposed. Also the results are influenced by the light conditions during experiments. Can the authors elaborate on the limitations of their current approach?

  • We completely agree on this point. The complexity and interconnectivity of traits, growth performance, nitrogen uptake and use are difficult to assess. We are aware that the trait of biomass under low nitrogen is strongly limited for statements about Nitrogen use efficiency. However, biomass is one of the most important traits for crop production. Further our manuscript’s emphasis is on the identification of the novel QTL. An explanatory sentence was added in line 355.

Authors need to better justify in the introduction and discussion how the approach used and the findings from this study can advance development of crops with increased nitrogen use efficiency.

  • A new sentence was added in the beginning of 4.3.

The authors emphasize discussion of plant color, which was not a trait measured or presented in the article. This should be avoided and authors should limit their conclusions and discussion to data obtained in their study.

  • We completely agree on this point. As the trait of color change was not analyzed in detail, we removed this section. The correlation of anthocyanin accumulation and leaf coloring will be analyzed in detail in future experiments.

Specific comments:

Line 34 – 49. This introduction is too general. Please rewrite to justify the approach in the present paper, and not the need to improve NUE in general.

Introduction was shortened and partially rewritten to focus on our approach.

Line 50 – Replace “the biomass” with “biomass”

  • Replaced (now line 57)

Line 70 – accessions were grown in either agar or soil, or under both conditions?

  • Both conditions, sentence changed

 Line 83 to 85 – Justify why these insertions were selected

  • The lines were selected to induce changes in DNA methylation patterns in the QTL region. explanation added in 3.5.

Line 85 – 86 – not clear weather authors selected accessions with different methylation patters as part of the work in this study or previous work

  • This is novel work and the accessions were selected based on available information about the DNA methylation pattern. Sentence in line 85 ff changed.

Line 90 – What selected lines?

  • More information added.

Line 95. Replace ‘12’ with ‘Twelve’.

  • replaced

Line 167-168 – Rewrite sentence for more clarity.

  • Sentence rephrased: The 102 accessions analysed previously [7] originate from different geographic regions (Figure 1). Plant biomass accumulation of accessions grown on low N in agar has been shown to vary considerably

Line 107 – Please provide additional details about the shoot data collected and in what treatments. Was it collected only under the low N? the reference cited says: “analysis of shoot growth of the populations on low N soil was performed according to Tschoep et al. (2009)”. Right now readers need to go to an additional source to understand the methods.

Sentence changed to: Shoot biomass data for GWAS of all accessions and mRILs was taken from [7], where the plants had been screened for biomass production in an agar plate-based high-throughput procedure with limited supply of nitrogen (0.05 mM KNO3).

Line 108 – replace “assessed’ by the precise measurements that were taken. How often, and in what treatments?

  • Sentence changed in: Root growth parameters were initially measured by a ruler and weighted from plants grown on vertical agar plates

Line 110 – 111 – What type of data was obtained from imaging? This data is not presented in the article. Present the data or remove these protocols from the methods.

  • Part was removed.

Line 155 – 157 – Describe what were the factors considered in the ANOVA and if they were considered fixed or random. How were the replicated experiments treated in the model?

Amended to: Adjusted means of phenotypic data were estimated by a 2-factor ANOVA with ‘accession’ and ‘nitrogen’ as main factors (accession + nitrogen + accession.nitrogen), and ‘experiment’ as blocking factor. Adjusted means of qPCR expression data were estimated by a simple ANOVA with ‘genotype’ as main factor and ‘biological replicate’ as blocking factor. Significant differences were determined after Tukey multiple testing correction at 5%.

Line 166 – Replace “plant performance” by other traits or processes that were affected in the study cited.

  • Replaced with biomass accumulation

Line 167-168 – Rewrite sentence for more clarity.

  • Sentence rephrased: The 102 accessions analysed previously [7] originate from different geographic regions (Figure 1). Plant biomass accumulation of accessions grown on low N in agar has been shown to vary considerably

Line 253 – Not clear if the accessions selected were selected because of being superior, or because of their different methylation patterns in the three candidate genes.

  • Sentence edited: After identification of the aforementioned QTL, the influence of DNA methylation pattern was tested. Therefore accession with different methylation pattern were selected and analysed more in detail using a novel experimental setup.

Line 254 – the authors seem to be presenting results from another paper. The text must be more clear indicating what is part of this paper, or from previous work.

  • We tried to make this point more clear: The raw data used for GWAS was generated in prior work. The GWAS based on these data is new.

Line 261 – throughout the article there is inconsistency in naming biomass, biomass mass, fresh weight, shoot mass,  ‘Biomass mass’ is incorrect. Use shoot fresh mass throughout the article, biomass usually refers to dry weight, and on an area basis.

  • Corrected

Line 261 – What is maximal root extension? Root length? Use appropriate and specific names to define traits measured.

  • Traits were defined, abbreviations explained,

Line 267 – Replace   “decrease in shoot biomass of growth resulting in the smallest plant” by “decrease in shoot fresh mass”

  • Corrected

Figure 6  and associated text in results section – Statistics from the ANOVA should be provided in the Figure, and mentioned in the presentation of results. Avoid presenting results and tendencies and focus on the statistical analysis (e.g. Line 267-268). Results from root data are not presented in the results section.

Paragraph changed to: Rosette biomass (Fresh weight: FW, Figure 6A) and root length of the main root (Root length: RL, Figure 6B) were estimated 44 days after sowing at the end of the experiment, using 2-factor ANOVA. We found highly significant differences (p<0.001) for all three factors (accession, nitrogen and the interaction term accession.nitrogen) for fresh weight, indicating differential growth reactions of the accessions to limited nitrogen (Table S6 and Figure 6A). Based on the biomass data, Appt-1 and Gy-0 were considered good performers and Col-0 and Cvi-0 bad performers. As general response to nitrogen deficiency, Arabidopsis plants have been described to develop longer roots [35]. We found highly significant differences (p<0.001) for accession and the interaction term accession.nitrogen) for root length (Table S6 and Figure 6B), but only Col-0 and Cvi-0 displayed longer roots under low N than sufficient N.

Line 361 – this the first mention that Col-0 performs poorly under N limiting conditions. Poorly relative to other accessions? Where is this data presented or discussed. There is not enough context given.

  • This was already based on Figure 6/7 and explained in 279ff.

Line 326-327 – What is a reduced stress response?

  • Sentence removed.

Line 327-328 – the color differences are not apparent in Figure 8.

  • We agree that the pictures do not properly support the statement. As the trait of color change was not analyzed in detail, we removed this section. The correlation of anthocyanin accumulation and leaf coloring will be analyzed in detail in future experiments.

Line 367 – the color differences are not apparent in Figure 5. Why is data obtained from imaging not used to obtained quantitative results, instead of relying on a subjective interpretation of color?

  • We agree that the pictures do not properly support the statement. As the trait of color change was not analyzed in detail, we removed this section. The correlation of anthocyanin accumulation and leaf coloring will be analyzed in detail in future experiments.

Line 374-375 – This paper does not: 1) analyze how production of anthocyanin relates with biomass accumulation, and 2) measure in any way red color (or at least this data is not presented). Authors should not make any interpretation or conclusion based on color or anthocyanin accumulation, or if it is independent or not from biomass accumulation.

  • As the trait of color change was not analyzed in detail, we removed this section. The correlation of anthocyanin accumulation and leaf coloring will be analyzed in detail in future experiments.

We again thank all reviewers for the constructive criticism. We hope that we could address all raised points and improve the manuscript for final acceptance in Agronomy.

Yours Sincerely,

Markus Kuhlmann